# The effects of chikungunya virus infection on people living with HIV during the 2014 Martinique outbreak

**Mathilde Pircher[1], Edwin Pitono[1], Sandrine Pierre-François[1], Sabine Molcard[2], Lauren Brunier-Agot[3], Laurence Fagour[4,5], Fatiha Najioullah[4,5], Raymond Cesaire[4,5], Sylvie Abel[1,5], Lise Cuzin[1,6]*, André Cabié[1,5,7]**

1 Infectious Diseases and Tropical Medicine Unit, University Hospital of Martinique, Fort de France, France, 2 Unit of Physical and Rehabilitation Medicine, University Hospital of Martinique, Fort de France, France, 3 Department of Rheumatology, University Hospital of Martinique, Fort de France, France, 4 Virology Laboratory, University Hospital of Martinique, Fort de France, France, 5 Antilles University, EA 7524, Fort-de-France, France, 6 INSERM UMR1017, Toulouse III University, Toulouse, France, 7 INSERM CIC1424, Fort-de-France, France

* lise.cuzin@chu-martinique.fr

**Data Availability Statement:** All relevant data are within the manuscript and its Supporting Information files.

**Funding:** The authors recieved no specific fundingfor this work.

## Abstract

Our objective was to describe the clinical presentation of chikungunya virus (CHIKV) infection in patients living with HIV (PLHIV) during the 2014 Martinique outbreak. During the outbreak and the 6 following months, all PLHIV coming in our unit for a medical evaluation answered questions about potential CHIKV related symptoms, and had blood tests to assess the diagnosis. For patients coming in at the acute phase of infection, we are able to provide and analyze CD4+, CD8+ T-cells and HIV viral load evolution before, during and after CHIK infection. Among the 1 003 PLHIV in care in the center at the time of the outbreak, 188 (94 men and 94 women) had confirmed (following the WHO definition) CHIKV infection. Clinical presentation was common in 63% of the cases, severe and atypical forms were scarce. During the acute phase, CD4+ and CD8+ T-cells (evaluated in 30 PLHIV, 15 men and 15 women) absolute numbers dropped significantly, but returned to pre-CHIKV values after the acute phase. Reassuringly, CD4 and CD8 T cells proportions did not decrease during the acute phase. CHIKV infection had no significant impact on this anti-retroviral treated population.

## Introduction

Chikungunya virus (CHIKV) is an alphavirus of the *Togaviridae* family, transmitted to humans by Aedes mosquitoes. The Martinique and Guadeloupe 2014 outbreak started in December 2013, lasted till December 2014 and affected about 40% of the population [1]. The effects of CHIKV infection in people living with HIV (PLHIV) have rarely been described, despite the fact that these infections share common geographic distributions [2, 3]. The objective of this study was to describe the clinical manifestations of confirmed symptomatic CHIKV infection in PLHIV living in Martinique and the potential impact of chikungunya on HIV infection.

**Competing interests:** The authors have declared that no competing interests exist.

## Material and methods

### Background information

Martinique is an overseas territory of the French West Indies. This island of 11 00 km$^2$ has a population of 378 243 inhabitants; 19% are <15 years old, 59% between 15 and 60 years old, and 22% >60 years old. The University Hospital of Martinique (UHM) is a tertiary referral center providing healthcare, teaching, and research programs. The Infectious Diseases and Tropical Medicine Unit of UHM is the only specialized resource for PLHIV in Martinique, thus almost all the 1000 PLHIV living on the island in 2014 visited the unit at least once a year. Briefly, among PLHIV in Martinique in care in 2014, 63% were men, median age was 49 years (from 15 to 80), and the most frequent way of acquisition was sexual (95%).

### Study population

All patients coming for a clinical visit between December 2013 and June 2015 (thus covering the outbreak and the next 6 months) were questioned about potentially acute chikungunya related symptoms [4]. All the PLHIV who had had clinical CHIKV manifestations (at least history of fever or acute onset joint pain) during the outbreak had biological assay for CHIKV infection confirmation. WHO defines CHIKV cases as "confirmed" if patients have biological confirmation regardless of clinical presentation [5]. In this definition, biological confirmation is based on reverse transcriptase polymerase chain reaction (RT-PCR) (positive during 7 days after the onset of symptoms) and/or serology (IgM appearing around day 4 and disappearing before day 90, and IgG present from day 10). Thus, the confirmation method used in our study varied following the delay between symptoms' onset and the date of clinical visit.

PLHIV who had had clinical CHIKV manifestations during the outbreak with biological confirmation at the time of their visit in the Infectious Diseases and Tropical Medicine Unit were included in the study. Patients who had had clinical CHIKV manifestations during the outbreak but before HIV diagnosis were excluded.

All patients with confirmed CHIKV infection were classified according to clinical form for the acute phase of CHIKV infection described by the WHO [4]. Patients were classified as having acute clinical infection if they had fever of 38.5˚C or higher, acute onset joint pain, and RT-PCR positive for CHIKV. Exanthema, myalgia, back pain, headache, vomiting, and diarrhea without effect on overall general health status, were considered to be signs and symptoms usually accompanying the typical acute phase course of CHIKV infection. Patients were classified as having atypical infection if they had fever of 38.5˚C or higher, acute-onset joint pain, and other manifestations such as neurological, cardiovascular, dermatological, ophthalmological, hepatic, renal, respiratory or hematological signs, decompensated diabetes mellitus, balance or walking disturbances, or concomitant infections. Patients were classified as having severe acute infection if they had fever of 38.5˚C or higher, acute onset joint pain, and at least one organ or system failure that was life-threatening and required hospitalization. Patients without fever or acute-onset joint pain, or with missing data were considered to have unclassifiable infection. Patients with previous clinical diagnosis of chikungunya, with a positive chikungunya laboratory test, presenting with at least one of the following articular manifestations: pain, rigidity, or edema, continuously or recurrently after 12 weeks of the onset of the symptoms were classified as chronic form of chikungunya.

### Data collection and ethics statement

The Dat'AIDS cohort (Clinicaltrials.gov reference NCT02898987) was started in 2000, including at that time all PLHIV with their previous medical history, and all new patients in care

since then have become part of the cohort, after being suitably informed and providing written consent [6]. CHIKV infection clinical signs were prospectively added to the routinely collected information after additional patients' information and consent. For PLHIV assessed during the acute CHIKV infection, we are thus able to analyze CD4 and CD8 T cell counts as well as HIV viral load before, during and after acute CHIKV infection. Data were extracted and fully anonymized for analysis on December, 1st, 2017. The database is available as supporting information (S1 Dataset).

## Statistical analysis

Analysis was performed using Stata software version 12 (StataCorp LP, College Station, TX, USA). Categorical variables were summarized using frequencies and percentages and compared using Fischer-Exact test. Continuous variables were summarized using median, first and third quartiles [Q1-Q3)], and compared using non-parametric tests (Mann-Whitney test or

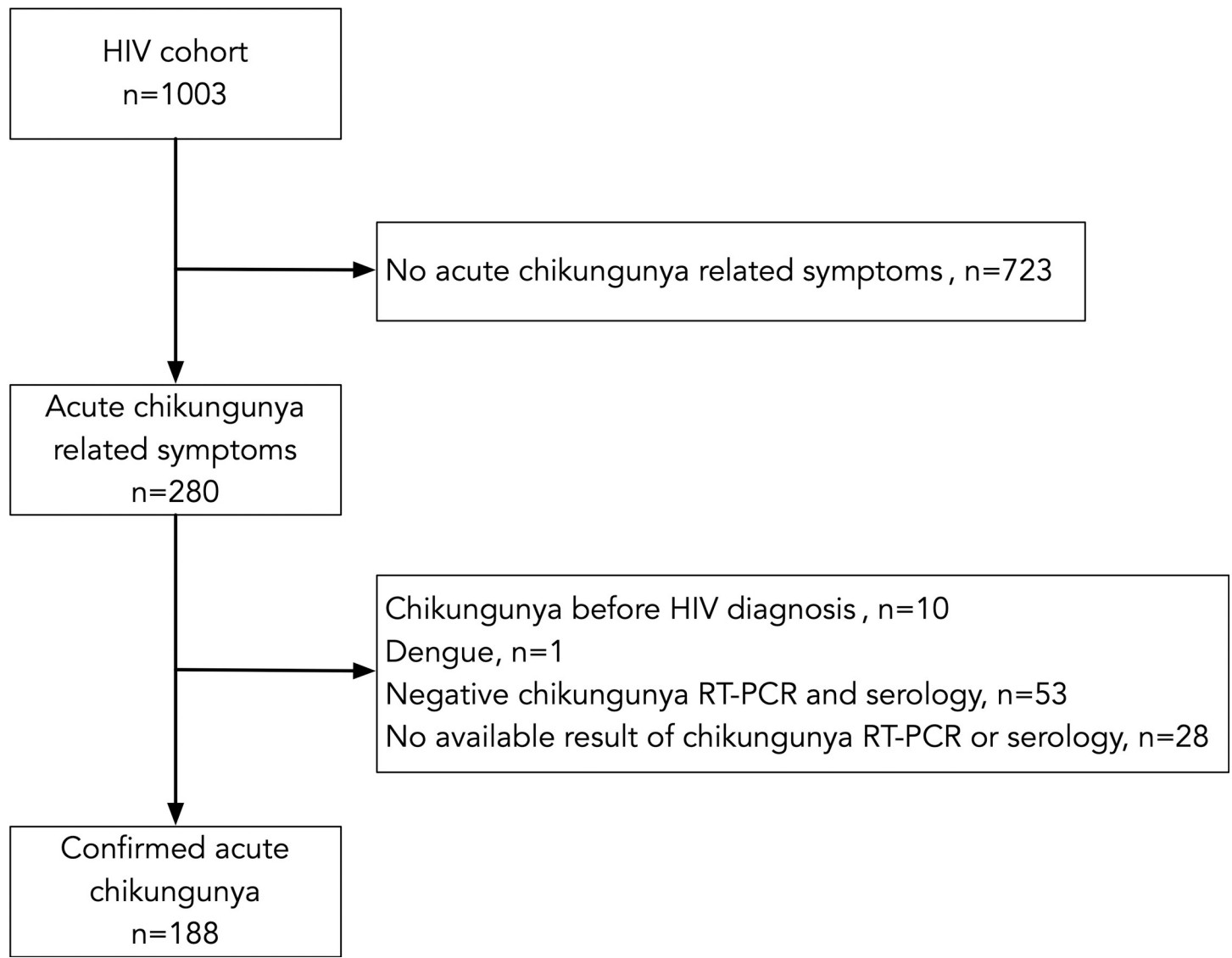

**Fig 1. Enrolment and follow-up of the study participants.**

Wilcoxon matched-pairs signed-rank test as appropriate). A P value of less than 0.05 was considered statistically significant.

## Results

Among 1,003 PLHIV in care in 2015, 188 (94 men and 94 women) had a clinical and further confirmed CHIKV infection (Fig 1). The median age of the patients was 49.7 years [42.5–59.3], ranging from 19 to 86.

### Clinical manifestations of chikungunya in PLHIV

In 62.8% of the population the presentation was common (Table 1). Chronic forms were described in 21.8% of our cases. Atypical forms were described in 2 patients (thrombocytopenia < 50 G/L without bleeding, alanine aminotransferase (ALT) greater than 10 times the upper limit of normal without criteria for acute liver failure). Severe forms were described in 4 patients: congestive heart failure (2 cases), encephalopathy (1 case) and acute renal failure (1 case). No death has been reported in relation with CHIKV.

Biological confirmation was made by RT-PCR in 37 (19.6%) cases, or by serological test in 151 (80.4%). The median time between clinical manifestations and the visit during which the patients were clinically and biologically assessed was 6.1 months [1.6–10.3].

### Impact of chikungunya on HIV infection

In our anti-retroviral treated patients, HIV viral load was below detection (50 copies/ml) in 86.7%, with no changes during acute CHIKV infection. No clinical HIV progression has been reported in relation with CHIKV in our population.

An HIV visit occurred in 30 cases during the acute CHIKV infection. Lymphocytes, CD4 T-cells and CD8 T-cells counts, as well as viral load assessment were available before (median delay 12.6 weeks [4.3–22.3]), during and after (median delay 7.8 weeks [2.9–16.6]) acute CHIKV infection. Median CD4+ T-cells count was $721.3/mm^3$ before, $428.2/mm^3$ during (p<0.0001 by comparison to the value before CHIKV infection) and $726.8/mm^3$ after the acute phase, while there was no change in CD4 percentages. The same differences have been observed in the CD8+ T-cells counts and in the total lymphocytes counts (Fig 2 and Table 2).

**Table 1. Characteristics of CHIKV related symptoms in 188 PLHIV.**

| Clinical characteristics | | N (%) |
|---|---|---|
| Clinical forms | Common | 118 (62.8) |
| | Atypical | 2 (1.1) |
| | Severe | 4 (2.3) |
| | Unclassifiable | 64 (34.0) |
| Symptoms | Fever (n = 162) | 130 (80.3) |
| | Arthralgia (n = 164) | 158 (96.3) |
| | Myalgia (n = 161) | 51 (31.7) |
| | Headaches (n = 161) | 54 (33.5) |
| | Skin rash (n = 161) | 52 (32.3) |
| | Other symptoms (n = 161) | 43 (26.7) |
| | Digestive symptoms (n = 161) | 24 (14.9) |
| Evolution | Hospitalization | 9 (5.2) |
| | Chronic forms | 41 (21.8) |

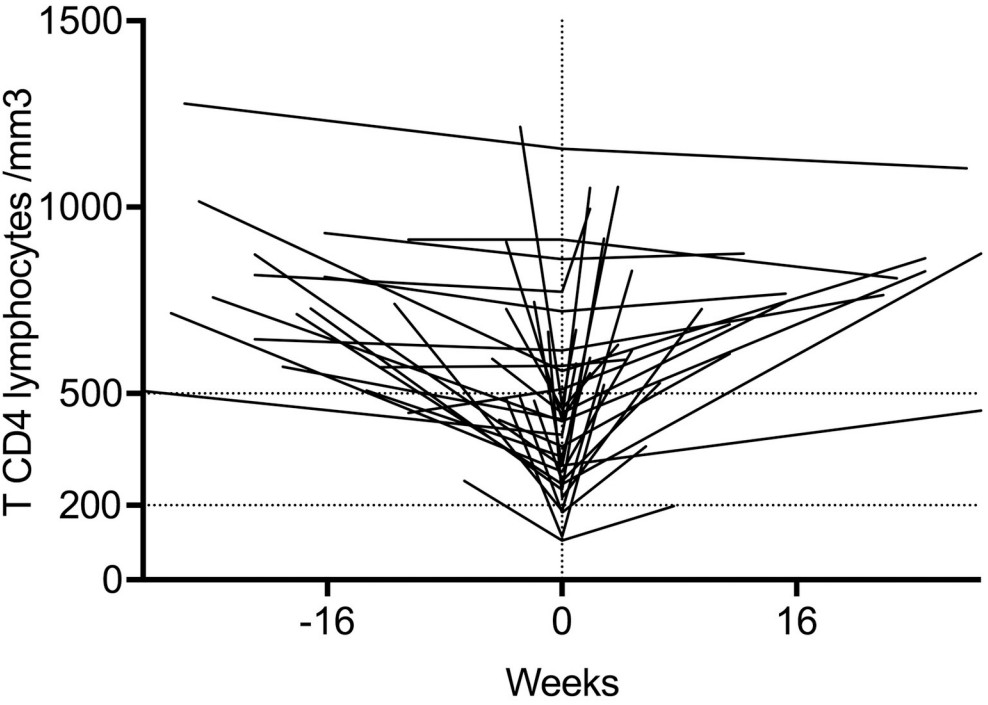

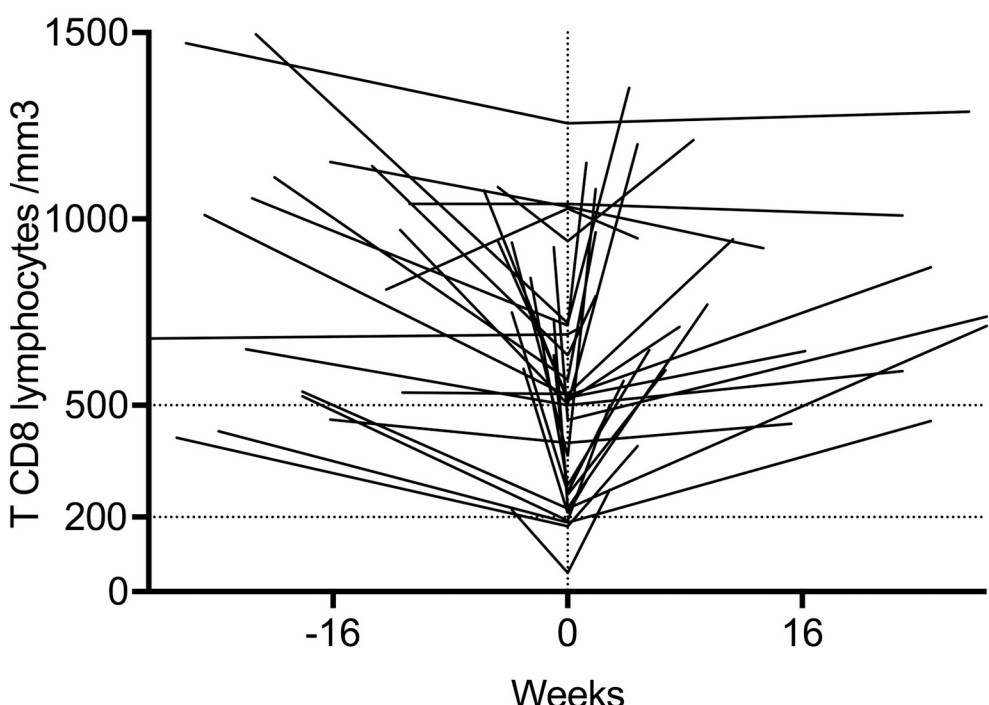

**Fig 2. CD4+ and CD8+ T-cells evolution (absolute numbers) during CHIKV infection (Week 0: Acute CHIKV infection).**

**Table 2. Lymphocytes, CD4 and CD8 T-cells count (absolute number and %), and CD4/CD8 ratio, prior, during and after acute chikungunya infection.**

|  | Prior [a] | During | After [b] | P [c] |
|---|---|---|---|---|
| Lymphocytes (cells/mm³) | 2060 [1683–2520] | 1045 [680–1685] | 1896 [1528–2480] | 0.0001 |
| CD4 T-cells, (cells/mm³) | 721.3 [566.7–873.9] | 428.2 [266.2–574.1] | 736.5 [589.7–875.2] | 0.0001 |
| CD4 T-cells, % | 32.1 [27.7–38.6] | 34.4 [29.1–42.5] | 33.5 [29.2–41.6] | 0.34 |
| CD8 T-cells (cells/mm³) | 883.2 [598.0–1077.0] | 511.5 [264.4–717.0] | 781.5 [592.0–1009.0] | 0.0001 |
| CD8 T-cells, % | 38.2 [29.8–45.5] | 36.9 [29.4–45.0] | 36.7 [29.8–43.9] | 0.18 |
| CD4/CD8 ratio | 0.8 [0.6–1.0] | 1.0 [0.7–1.2] | 0.9 [0.7–1.2] | 0.12 |

[a]median delay 12.6 weeks [4.3–22.3], [b]median delay 7.8 weeks [2.9–16.6], [c]comparison between "prior" and "during" acute chikungunya infection

## Discussion

To the best of our knowledge, we provide the first description of clinical manifestations of CHIKV infection in PLHIV. The proportions of non-typical and severe forms in our population seem higher than those reported in the Reunion Island's 2005 outbreak, in which 0.2% were non-atypical and 0.1% severe [7]. Notably we included only adult PLHIV, that differ regarding gender and age form the general Martinique population (S1 Table), while they analyzed a more global population, including pregnant women and children.

Evolution to chronic manifestations was described in 22% of our population. Evolution to chronic symptomatology has been described in 40% of the patients seen by general practitioners in relation with an acute CHIKV infection during the Caribbean outbreak [8]. This proportion was much higher after the Reunion Island outbreak [9] and in the global America's 2014 outbreak [10]. Because the median time between acute phase and our evaluation was 6 months, a memory bias is possible. Nevertheless, because of the debilitating evolution of chronic forms we do think that this bias is of small importance. Chronic forms are more frequent in ageing patients [9, 11, 12], thus our study population may be too young to describe these forms.

The CHIKV related lymphopenia has already been described [2, 13, 14]. The absence of modification in proportions of CD4+ and CD8+ T-cells and the restauration of pre-CHIKV values in the first months after acute infection are reassuring. Furthermore, and as in a previous study [3], CHIKV infection has not been responsible for virological rebound, although our population may be too small to observe such an effect.

Although this retrospective study provides reassuring information regarding CHIKV infection in PLHIV, it clearly has limits. As stated, memory bias cannot be excluded in our study, due to the fact that a large part of the population was included after their acute CHIKV infection. Another limit is that we only performed biological confirmation in patients who presented with or remembered having had clinical manifestations, but as CHIKV infections are mostly symptomatic [10] this bias may be of small importance.

In conclusion, while CHIKV infection is a public health problem because of the debilitating chronic forms, we describe the absence of severe evolution of CHIKV in PLHIV. CHIKV induced a transient absolute lymphopenia with no clinical consequence, without change of CD4+ and CD8+ T cells proportions, and without any virological impact in treated patients.

## Supporting information

**S1 Dataset.**
(XLSX)

**S1 Table.**
(DOCX)

## Author Contributions

**Conceptualization:** Sylvie Abel, André Cabié.

**Data curation:** Mathilde Pircher, André Cabié.

**Investigation:** Edwin Pitono, Sandrine Pierre-François, Laurence Fagour, Fatiha Najioullah.

**Methodology:** Mathilde Pircher, Edwin Pitono, Sabine Molcard.

**Project administration:** Mathilde Pircher, André Cabié.

**Supervision:** Mathilde Pircher, Sandrine Pierre-François, Sabine Molcard, Lauren Brunier-Agot, Fatiha Najioullah, Raymond Cesaire, André Cabié.

**Validation:** Lauren Brunier-Agot, Laurence Fagour, Fatiha Najioullah, André Cabié.

**Writing – original draft:** Mathilde Pircher, Lise Cuzin.

**Writing – review & editing:** Raymond Cesaire, Sylvie Abel, Lise Cuzin, André Cabié.

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
