## [Decision Letter · Decision Letter 0]

8 Apr 2020

PONE-D-20-04293

The Effects of Chikungunya Virus Infection on People Living with HIV during the 2014 Martinique outbreak.

PLOS ONE

Dear Dr. Cuzin,

Thank you for submitting your manuscript to PLOS ONE. After careful consideration, we feel that it has merit but does not fully meet PLOS ONE’s publication criteria as it currently stands. Therefore, we invite you to submit a revised version of the manuscript that addresses the points raised during the review process.

they remained quite minor but as noted, it should be important to highlight the current situation of HIV prevalence in the study period and within a comparable population as requested by the reviewer.

We would appreciate receiving your revised manuscript by May 23 2020 11:59PM. To enhance the reproducibility of your results, we recommend that if applicable you deposit your laboratory protocols in protocols.io, where a protocol can be assigned its own identifier (DOI) such that it can be cited independently in the future. For instructions see: http://journals.plos.org/plosone/s/submission-guidelines#loc-laboratory-protocols

We look forward to receiving your revised manuscript.

Kind regards,

Pierre Roques, Ph.D.

Academic Editor

PLOS ONE

Journal Requirements:

2. Please provide additional details regarding participant consent. In the ethics statement in the Methods and online submission information, please ensure that you have specified whether consent was suitably informed.

3. Please include additional information regarding the survey or questionnaire used in the study and ensure that you have provided sufficient details that others could replicate the analyses. If you developed and/or translated a questionnaire as part of this study and it is not under a copyright license more restrictive than Creative Commons Attribution (CC-BY), please include a copy, in both the original language and English, as Supporting Information.

4. Please ensure that you refer to Figure 2 in your text as, if accepted, production will need this reference to link the reader to the figure.

Reviewers' comments:

Reviewer's Responses to Questions

**Comments to the Author**

1. Is the manuscript technically sound, and do the data support the conclusions?

Reviewer #1: Yes

2. Has the statistical analysis been performed appropriately and rigorously? 

Reviewer #1: Yes

3. Have the authors made all data underlying the findings in their manuscript fully available?

Reviewer #1: Yes

4. Is the manuscript presented in an intelligible fashion and written in standard English?

Reviewer #1: Yes

5. Review Comments to the Author

Reviewer #1: The relationship between arbovirus infections and HIV is understudied. This manuscript provides a much needed view to the effect of CHIKV infections on HIV patients. A population that during outbreaks will require most attention. That being said, the manuscript requires further editorial attention.

1) the numbering system used needs to be adjusted to the US numbering format.

2) Line 121, please describe that the virus load needed is the HIV.

3) in describing population , please include the upper and lower limit of the age group not just the median age.

5) A general description for the level of CHIKV infection in Martinique during the outbreak would be useful.

6) can you please include a timeline for when the questionnaires of patients were taken post the outbreak? this would help asses how much of recall bias is to be expected.

6. PLOS authors have the option to publish the peer review history of their article (what does this mean?). If published, this will include your full peer review and any attached files.

Reviewer #1: No

---

## [Author Response · Author response to Decision Letter 0]

5 May 2020

Dear Editor

Thank you for the time you and the reviewer took to assess our manuscript. We did our best to answer the wanted revisions.

Editor queries

1. It should be important to highlight the current situation of HIV prevalence in the study period 

Answer: Some details have been added in the Background paragraph

Answer: We did our best

2. Please provide additional details regarding participant consent. In the ethics statement in the Methods and online submission information, please ensure that you have specified whether consent was suitably informed.

Answer: Details have been added in the data collection and ethics statement part  3. Please include additional information regarding the survey or questionnaire used in the study and ensure that you have provided sufficient details that others could replicate the analyses. If you developed and/or translated a questionnaire as part of this study and it is not under a copyright license more restrictive than Creative Commons Attribution (CC-BY), please include a copy, in both the original language and English, as Supporting Information.

Answer: As we understand that the formulation led to misunderstanding, I rephrased the sentence, line 58 of the Manuscript with track changes

4. Please ensure that you refer to Figure 2 in your text as, if accepted, production will need this reference to link the reader to the figure.

 Answer: Edited

Reviewer’s queries

1) the numbering system used needs to be adjusted to the US numbering format.

Answer: I did my best 2) Line 121, please describe that the virus load needed is the HIV.

Answer: Edited 3) in describing population, please include the upper and lower limit of the age group not just the median age.

Answer: I did, although quartiles may be more informative, thus I propose to keep both

5) A general description for the level of CHIKV infection in Martinique during the outbreak would be useful.

Answer: As stated in the Introduction, CHIKV infection affected about 40% of the island population

6) can you please include a timeline for when the questionnaires of patients were taken post the outbreak? this would help assess how much of recall bias is to be expected

Answer: Precision has been added (L123 and 124 of the revised manuscript with track changes) regarding the time elapsed between clinical manifestations, assessment of symptoms and biological confirmation for the overall population.

---

## [Editor Report · Decision Letter 1]

22 May 2020

The Effects of Chikungunya Virus Infection on People Living with HIV during the 2014 Martinique outbreak.

PONE-D-20-04293R1

Dear Dr. Cuzin,

We are pleased to inform you that your manuscript has been judged scientifically suitable for publication and will be formally accepted for publication once it complies with all outstanding technical requirements.

With kind regards,

Pierre Roques, Ph.D.

Academic Editor

PLOS ONE
---

## [Editor Report · Acceptance letter]

27 May 2020

PONE-D-20-04293R1 

The Effects of Chikungunya Virus Infection on People Living with HIV during the 2014 Martinique outbreak. 

Dear Dr. Cuzin:

I am pleased to inform you that your manuscript has been deemed suitable for publication in PLOS ONE. Congratulations! Your manuscript is now with our production department. 

With kind regards,

on behalf of

Dr. Pierre Roques 

Academic Editor

PLOS ONE